# Effects of Environmental Factors on the Diversity of Grasshopper Communities along Altitude Gradients in Xizang, China

**DOI:** 10.3390/insects15090671

**Published:** 2024-09-04

**Authors:** Yonghui Li, Qing Liu, Xiaoming Zhang, Benyong Mao, Guohui Yang, Fuming Shi, Jingui Bi, Zhibin Ma, Guowen Tang

**Affiliations:** 1Research Institute of Gaoligong Mountains, Baoshan University, Key Laboratory of Conservation and Utilization of Insect Resources in Western Yunnan, Baoshan 678000, China; liyonghui0705@163.com (Y.L.); liuqinggc065@126.com (Q.L.); bijingui2022@163.com (J.B.); m18288957220@163.com (Z.M.); 2Baoshan Key Laboratory of Biodiversity Conservation and Utilization of Gaoligong Mountains, Baoshan 678000, China; 3College of Plant Protection, Yunnan Agricultural University, National Key Laboratory for Conservation and Utilization of Biological Resources in Yunnan, Kunming 650201, China; zxmalex@126.com; 4College of Agriculture and Biological Science, Dali University, Dali 671003, China; maoby65@sohu.com (B.M.); yanggh727@sina.com (G.Y.); 5School of Life Science, Institute of Life Science and Green Development, Hebei University, Baoding 071002, China; shif_m@126.com

**Keywords:** altitudinal gradient, grasshopper community structure, distribution pattern, driving factors, Xizang

## Abstract

**Simple Summary:**

Environmental factors varying widely across altitudinal gradients contribute significantly to the intricate and diverse distribution patterns observed in insect communities. Investigating the diversity patterns in the grasshopper community along altitudinal gradients in Xizang is crucial for understanding broader trends in insect diversity. This study revealed a strong effect of altitude on grasshopper community diversity distribution in Xizang, showing that grasshopper species richness, Margalef richness index, Shannon–Wiener index, and Simpson dominance index decreased with an increase in altitude. The results of Pearson correlation analysis and hierarchical partitioning showed that temperature, moisture, and soil properties are closely related to the altitude distribution patterns of grasshopper communities. The key factors driving changes in grasshopper community diversity along altitudinal gradients include the mean annual temperature range, precipitation in the coldest season, and precipitation in the driest month. To summarize, the interplay between elevation and environmental variables significantly influences grasshopper community structure, distribution patterns, and diversity.

**Abstract:**

To determine the grasshopper species composition, altitudinal distribution patterns, and their main drivers, we conducted a study in Xizang using 33 sample plots ranging from 600 to 4100 m. Grasshoppers were collected from August to October during 2020–2022 using sweep nets. A total of 1159 grasshoppers from six families, 28 genera, and 44 species were identified, with *Omocestus cuonaensis* and *Aserratus eminifrontus* as the dominant species, comprising 30.03% and 10.26% of total grasshoppers, respectively. The results showed that species richness and the Margalef richness index of grasshopper communities decreased significantly (*p* < 0.05) with increasing altitude, peaking at 1100–1600 m and lowest values at 2600–3100 m. Similarly, the Shannon–Wiener index and Simpson dominance index also decreased significantly (*p* < 0.05) with an increase in altitude, showing the highest and lowest values at 600–1100 m and 3100–3600 m, respectively. The Jaccard similarity coefficients among grasshopper communities varied from 0 to 0.40 across altitudinal gradients, indicating different degrees of dissimilarity. The results of Pearson correlation analyses showed that the Shannon–Wiener index, species richness, Margalef richness index, and Simpson dominance index of grasshopper communities were significantly negatively correlated with the temperature factors and soil pH, but they were significantly positively correlated with the moisture factors. Hierarchical partitioning identified annual mean temperature–daily difference, precipitation in the coldest season, and driest month precipitation as the primary factors explaining variance in grasshopper community diversity in Xizang. These findings provided greater insights into the mechanisms underlying insect community structure, distribution patterns, and diversity in Xizang ecosystems, including implications for the effects of global warming on insect communities.

## 1. Introduction

Species diversity studies are essential for understanding community composition, maintaining structure, and ensuring stability [1]. Higher diversity often enhances ecosystem resilience and stability [2]. Grasshoppers have various habitats and are highly adaptable to the environment [3,4]. They occupy an important niche in the ecosystem and are good indicators of climatic conditions and vegetation structure on a fine scale [5]. High grasshopper densities can harm farmland and natural grassland ecosystems [6]. Thus, studying grasshopper diversity in specific regions is crucial for understanding species coexistence, supporting agricultural and ecological sustainability, and promoting biodiversity conservation [7].

The altitude difference in biodiversity is an important issue in ecological research [8]. As a comprehensive driving factor, altitude can lead to complex changes in environmental factors such as light, temperature, and precipitation, which are of great significance for studying the distribution pattern of biodiversity and its driving factors [9]. Dokjan et al. [10] showed that the species richness and abundance of Coleoptera insects were negatively correlated with altitude and temperature. Beirao et al. [11] observed that the altitude gradient would significantly affect the distribution pattern of butterflies, and the diversity of butterfly species was positively correlated with the temperature and humidity of the external environment within a certain range. Several studies have found that insect diversity and altitude gradients are predominantly influenced by monotonic decreases [12,13,14], monotonic increases [15,16], and mid-peak patterns, [10,17,18], influenced by altitude range and region. However, research on grasshopper species diversity primarily focuses on different types of habitat [19,20,21,22,23], disturbance conditions [23,24,25], and topography [26]. Few studies have investigated the diversity of grasshopper communities at different altitudes.

The distribution pattern and maintenance mechanism of insect species are core issues in insect ecology [27]. Several factors influence the elevation pattern of species diversity, including temperature, water availability, soil properties, plant diversity, and habitat heterogeneity [28,29]. Temperature and moisture play crucial roles in shaping insect diversity on a large scale. Temperature primarily affects insect metabolism and the energy required for growth and development. It varies systematically with altitude, thus affecting insect diversity distribution. Acharya and Vijayan [30] demonstrated the importance of temperature and water in butterfly species richness. Beirao et al. [11] observed that butterfly diversity increases with higher temperatures and greater water availability. Soil physical and chemical properties directly or indirectly affect the diversity of insects living on the soil surface. For example, a significant linear correlation was found between species richness and diversity indices of desert ground beetle communities and soil water content and organic matter [31]. Badenhausser et al. [32] found that more complex plant species support greater grasshopper community diversity and higher community dominance. However, due to variations in environmental parameters such as climate, topography, soil, and vegetation types across regions, achieving consensus on their effect on insect communities is challenging. Thus, further studies are needed to elucidate the mechanisms underlying the maintenance of insect community structure, distribution patterns, and diversity across different ecosystems.

Xizang, located in the southwest of the plateau, features complex terrain and unique natural and climatic conditions, making it a fragile and sensitive ecosystem ideal for studying how biodiversity is distributed with altitude [33]. Currently, no study has investigated the changes in grasshopper community structure, distribution patterns, and diversity along altitudinal gradients in Xizang. To provide more information on this topic, in this study, the sweep net method was used to survey grasshoppers within elevations ranging from 600 to 4100 m above sea level in Xizang. This study aims to clarify the composition and altitude distribution patterns of grasshopper communities in this area, thus investigating how grasshopper communities change along altitudinal gradients in response to environmental conditions. Ultimately, this study aims to provide basic information for the conservation of insect diversity in Xizang.

## 2. Materials and Methods

### 2.1. Overview of the Study Area

Xizang lies in southwestern Qinghai–Xizang Plateau, along China’s southwestern border, spanning from coordinates 78°25′~99°06′ E and 26°50′~36°53′ N. With an average altitude surpassing 4000 m, Xizang covers 1.202189 million km^2^. Its terrain slopes from northwest to southeast, characterized by prominent features like the Himalayas, the southern Xizang valley, the northern Xizang plateau, and the eastern Xizang alpine area. Moving from northwest to southeast, the plateau transitions from a cold, dry temperate climate to a warmer, more humid subtropical and finally tropical climate. The average annual temperature is 8 °C. The Himalayas and southeastern Xizang are critical for biodiversity conservation in the southwestern mountainous region, with altitude variations exceeding 8500 m, diverse landforms, climates, and abundant vegetation types [34].

### 2.2. Sample Point Setting and Sampling Method

During this study, grasshopper surveys and collections were conducted annually in Xizang from August to October 2020–2022. Sampling points were positioned every 150 m along altitudinal gradients, with adjustments made within a 50 m margin due to geographical and vegetation constraints. Each sampling point was selected based on habitat conditions and comprised three 10 m × 10 m quadrats that were at least 50 m apart. Within each quadrat, grasshoppers were collected by sweep net, with surveyors moving along both diagonals of the quadrat. Collected grasshopper samples were preserved in collection bottles containing 70% anhydrous ethanol. The longitude, latitude, elevation, and slope of each sample plot were established using a hand-held GPS locator and compass; species, cover, number of plants (clumps) and height within each grassland sample plot were determined. Grasshopper identification was carried out using the reference Orthoptera Species File (OSF) [35]. The collections were made in south central Xizang plateau in a narrow range of latitudes (28–31° N), where lower levels were warm and humid and higher levels were cold. Based on the five vegetation types observed in the area (Figure 1), the region was categorized into seven elevation gradients: 600~1000 m (setting 6 sample points), 1100~1600 m (setting 5 sample points), 1600~2100 m (setting 3 sample points), 2100~2600 m (setting 3 sample points), 2600~3100 m (setting 8 sample points), 3100~3600 m (setting 3 sample points), and 3600~4100 m (setting 5 sample points).

### 2.3. Environmental Data

#### 2.3.1. Data on Temperature and Precipitation

We extracted 19 environmental factors from the World Climate Database WorldClim v. 2.1 (http://www.worldclim.org (accessed on 10 December 2023)) [36]. The acquired data are meteorological observations from 1970 to 2000 with a spatial resolution of a 30-arc second. These factors included metrics such as the annual mean temperature (Bio1), mean diurnal range (Bio2), isothermality (Bio3), temperature seasonality (Bio4), max. temperature of warmest month (Bio5), min. temperature of coldest month (Bio6), temperature annual range (Bio7), mean temperature of wettest quarter (Bio8), mean temperature of driest quarter (Bio9), mean temperature of warmest quarter (Bio10), mean temperature of coldest quarter (Bio11), annual precipitation (Bio12), precipitation of wettest month (Bio13), precipitation of driest month (Bio14), precipitation seasonality (Bio15), precipitation of wettest quarter (Bio16), precipitation of driest quarter (Bio17), precipitation of warmest quarter (Bio18), and precipitation of coldest quarter (Bio19). Using the ‘raster’ package in R version 4.3.2, we extracted and analyzed these bioclimatic factors from the data maps.

#### 2.3.2. Vegetation Factor Data 

We obtained biological factors, a 30-arc second resolution grid is provided, which includes the Simpson index of vegetation (Zhsim), Shannon index of vegetation (Zhsha), evenness index of vegetation (Zheve), habitat homogeneity index (Tzh), and habitat heterogeneity index (Yzhi) from the EarthEnv—Habitat Heterogeneity website (http://www.earthenv.org/texture (accessed on 12 December 2023)) [37]. These five vegetation factors were extracted for further analysis from the data map using the ‘raster’ and ‘rgdal’ packages in R.

#### 2.3.3. Soil Factor Data

Soil factor data, including soil organic carbon (Soc), soil pH (Phh2o), soil total nitrogen (Nit), organic carbon density (Ocd), and soil sand ratio (Sand), were obtained from the National Earth System Science Data Center (http://www.geodata.cn (accessed on 14 December 2023)) [38]. Using the ‘raster’ and ‘geodata’ packages in R, we extracted the five soil factors from the data map based on the latitude and longitude coordinates at the sampling locations for further analysis; soil variables were estimated at a depth of 0.3 m.

### 2.4. Community Diversity Index

#### 2.4.1. Diversity Analysis of the Grasshopper Community

Species richness (S), Shannon–Wiener index (H), Pielou index (E), Margalef richness index (R), and Simpson dominance index (D) were used to assess species diversity and community characteristics [39,40,41]. Species richness: the number of species is directly represented by the number of grasshopper species investigated in the area; Shannon–Wiener index: H = −∑ pi ln pi, pi = Ni/N; Pielou index: E = H/lnS; Margalef richness index: R = (S − 1)/ln N; Simpson dominance index: D = 1 − ∑pi^2^. In the above formula, pi is the proportion of individuals of the i species, and Ni is the individual of the i species.

#### 2.4.2. Jaccard Similarity Analysis of Grasshopper Community

A similarity coefficient was used to assess similarities and differences among distinct altitudinal gradients. The community similarity index (Cs) was determined using the Jaccard similarity coefficient formula [42]:Cs = c/(a + b − c)
In the formula, a represents the number of species in community A, b represents the number of species in community B, and c represents the number of species shared by communities A and B. Cs is categorized as follows: highly dissimilar when Cs ranges from 0 to 0.25, moderately dissimilar when Cs ranges from 0.25 to 0.50, moderately similar when Cs ranges from 0.50 to 0.75, and highly similar when Cs ranges from 0.70 to 1.00 [42].

### 2.5. Data Processing

Calculations of the diversity index for different groups and altitude gradients were performed using the ‘vegan’ package in R version 4.3.2. The ‘ggplot2′ package was used to construct a box plot depicting grasshopper diversity across various altitudinal gradients. The analysis of the relationship between altitude and grasshopper diversity index was performed by the linear regression method with the ‘ggpubr’, ‘ggplot2’, and ‘geosphere’ software packages.

The R 4.3.2 software packages, such as ‘pheatmap’, ‘corrplot’, ‘ggplot2’, and ‘ggrepel’, were used to analyze correlations between environmental factors and the diversity index. A Pearson correlation heat map was generated. Environmental factors served as explanatory variables, while the biodiversity index served as the response variable. Since species diversity and environmental data usually have significant spatial autocorrelation, regression and correlation analysis tend to be more significant. In order to eliminate the influence of significant spatial autocorrelation on the significance of the results, the results of regression and correlation analysis are tested by the modified *t*-test [43,44]. Explanatory variables were maintained in their original form, and the biodiversity index was subjected to the Hellinger transformation for redundancy analysis (RDA). The ‘rdacca.hp’ package facilitated hierarchical segmentation to quantify the independent interpretation rate of each environmental factor on the altitude distribution pattern of grasshopper diversity [45]. In R 4.3.2, modified *t*-tests were performed using the modified.ttest function in the ‘SpatialPack’ package, and the ‘iNEXT’ package was used to plot sparse extrapolation curves for species.

## 3. Results

### 3.1. Composition of Grasshopper Community in Xizang

A total of 1159 grasshoppers were collected in Xizang with Gomphocerinae being the most common (534/1159 = 46.1%) followed by Oedipodinae (148 or 12.8%), Melanopinae (12.3%), and Pyrgomorphinae (10.8%), with the remaining seven subfamilies making up 18.0%. *Omocestus cuonaensis* and *Aserratus eminifrontus* were the dominant species, representing 30.03% and 10.26% of the total grasshoppers collected (Table 1 and Appendix A). 

The grasshoppers collected were in 2 families, 11 subfamilies, 28 genera and 44 species. Of these, 18 genera (64.3%) were represented by only one species, demonstrating a predominance of single-species genera among the grasshoppers we collected from Xizang. The ratio coefficient of genera to species was 0.64. Subfamilies with ratios lower than this included Gomphocerinae (0.20) and Oxyinae (0.50), highlighting their relatively abundant presence and significant role in fauna analysis in Xizang (Table 1).

### 3.2. Diversity Characteristics of Grasshopper Families in Xizang

The relative frequency of grasshoppers collected are given in Table 2, with Gomphocerinae most common and Eyprepocnemidinae least common. Margalef richness was highest in Oedipodinae followed by Gomphocerinae, Acridinae, Catantopinae, Pyrgomorphinae, Cyrtacanthacridinae, Melanoplinae, and Oxyinae. Species richness was highest in Oedipodinae followed by Gomphocerinae, Catantopinae = Acridinae, Pyrgomorphinae, Melanoplinae = Cyrtacanthacridinae = Oxyinae, and Coptacrinae = Eyprepocnemidinae = Spathosterninae. Shannon–Wiener index and Simpson dominance index were highest in Oedipodinae followed by Acridinae, Catantopinae, Pyrgomorphinae, Gomphocerinae, Oxyinae, Melanoplinae, and Cyrtacanthacridinae. Pielou index was highest in Oxyinae followed by Melanoplinae, Acridinae, Pyrgomorphinae, Catantopinae, Oedipodinae, Cyrtacanthacridinae, and Gomphocerinae (Table 2).

### 3.3. Altitude Distribution Pattern of Grasshopper Community Diversity in Xizang

Figure 2 is the sparse curve of grasshopper communities at different altitudes. The real line-part of the sparse curve is the curve obtained by the interpolation of the measured values of the number of individuals and the number of species, and the imaginary line-part is the curve obtained by the predicted value of the extension of the measured values of the number of individuals and the number of species. Figure 2 depicts that with the increased number of investigated insects, the number of species at altitudes of 600–4100 m gradually stabilized, indicating that the sampling of grasshopper communities at seven altitude gradients was sufficient (Figure 2).

Variations were found in the diversity of grasshopper communities across different altitudes in Xizang. Significant differences were observed in species richness (Figure 3A, F = 3.315, *p* = 0.0147), Margalef richness index (Figure 3C, F = 4.693, *p* = 0.0032), and Simpson dominance index (Figure 3E, F = 3.526, *p* = 0.0174) between the altitudinal gradients. However, no significant difference was recorded in the Shannon–Wiener index (Figure 3D, F = 2.326, *p* = 0.0771) and Pielou index (Figure 3F, F = 1.601, *p* = 0.211) among these groups. Species richness and the Margalef richness index exhibited a decreasing pattern with increasing altitude (Figure 4, *p* < 0.05), reaching a maximum at 1100–1600 m and a minimum at 2600–3100 m (Figure 3A,C). Similarly, the Shannon–Wiener index and Simpson dominance index decreased with altitude (Figure 4, *p* < 0.05), peaking and exhibiting their lowest values in the range 600–1100 m and 3100–3600 m, respectively. The Pielou index and the number of individuals showed varying changes along the altitude gradient without a distinct trend (Figure 4, *p* > 0.05). The Pielou index reached its highest and lowest values at 2600–3100 m and 3100–3600 m, while the number of individuals reached its highest and lowest values at 2600–3100 m and 600–1100 m (Figure 3 and Figure 4).

### 3.4. Similarity Analysis of the Grasshopper Community at Different Altitudes in Xizang

The cluster analysis heat map revealed that grasshopper groups in Xizang were segregated into two categories along the altitudinal gradient. Altitudes of 600–1100 m and 1100–1600 m are classified as one category; altitudes of 1600–2100 m, 2100–2600 m, 3600–4100 m, 2600–3100 m, and 3100–3600 m are classified as one category. Figure 5 shows that Catantopinae, Coptacrinae, and Eyprepocnemidinae were the dominant groups at altitude 600–1100 m; Spathosterninae and Acridinae were the dominant groups at altitude 1100–1600 m; Gomphocerinae and Pyrgomorphinae were the dominant groups at altitude 2600–3100 m; Oxyinae and Cyrtacanthacridinae were the dominant groups at altitude 2100–2600 m; Oedipodinae was the dominant group at altitude 3600–4100 m; and Melanoplinae was the dominant group at altitude 1600–2100 m. These results indicated that altitude significantly influenced the distribution of grasshopper groups (Figure 5).

The Jaccard similarity analysis of grasshopper communities across seven altitude gradients in Xizang revealed coefficients ranging from 0 to 0.40, indicating either extreme or moderate levels of dissimilarity. The highest similarity coefficient of 0.40 was observed between the adjacent 2600–3100 m and 3100–3600 m altitudes, but even this level demonstrated moderate dissimilarity among grasshoppers present at these altitudes. Highly dissimilar relationships of 0.14 to 0.24 were observed between the four other altitudes, but 8 of the 21 relationships had no overlap in species at all, reflecting the overall dissimilarity of grasshoppers at various altitudes (Table 3).

### 3.5. Correlation between Grasshopper Community Diversity and Environmental Factors in Xizang

Six dependent variables (Shannon-Wiener index (H), species richness (S), Simpson dominance index (D), Pielou index (E) and the number of individuals (N) were correlated with 32 independent variables (Table 4). Significant negative correlations were found between the Shannon–Wiener index of the grasshopper community in Xizang and factors such as soil pH, mean diurnal range, temperature seasonality, and temperature annual range (*p* < 0.05). Conversely, significant positive correlations were observed between the Shannon–Wiener index and precipitation of driest month, precipitation of driest quarter, and precipitation of coldest quarter (*p* < 0.05). Simpson dominance index showed significant negative correlations with soil pH, mean diurnal range, temperature seasonality, and temperature annual range (*p* < 0.05), and significant positive correlation with annual precipitation, precipitation of wettest month, precipitation of driest month, precipitation of wettest quarter, precipitation of driest quarter, precipitation of warmest quarter, and precipitation of coldest quarter (*p* < 0.05). Species richness showed significant negative correlations with soil pH, mean diurnal range, temperature seasonality, temperature annual range, and precipitation seasonality (*p* < 0.05), and significant positive correlation with precipitation of driest month, precipitation of driest quarter, and precipitation of coldest quarter (*p* < 0.05). The individual number demonstrated significant negative correlations with annual precipitation, precipitation of wettest month, annual mean temperature, min. temperature of coldest month, mean temperature of driest quarter, mean temperature of coldest quarter, precipitation of wettest quarter, precipitation of driest quarter, precipitation of warmest quarter, and precipitation of coldest quarter (*p* < 0.05), while showing a significant positive correlation with mean diurnal range, isothermality, and organic carbon density (*p* < 0.05). The Margalef richness index was significantly negatively correlated with precipitation seasonality, mean diurnal range, temperature seasonality, temperature annual range, and soil pH (*p* < 0.05), and it showed a significant positive correlation with annual precipitation, precipitation of wettest month, precipitation of driest month, precipitation of driest quarter, and precipitation of coldest quarter (*p* < 0.05). No significant correlation was found between the Pielou index and any environmental factors (*p* > 0.05) (Figure 6).

### 3.6. Analysis of the Driving Factors of the Grasshopper Community Diversity in Xizang

This study considered 17 environmental variables, including Bio1 (annual mean temperature), Bio2 (mean diurnal range), Bio4 (temperature seasonality), Bio6 (min. temperature of coldest month), Bio12 (annual precipitation), Bio14 (precipitation of driest month), Bio16 (precipitation of wettest quarter), Bio17 (precipitation of driest quarter), Bio19 (precipitation of coldest quarter), Zhengbei evenness (vegetation evenness index), Zhengbei Shannon (vegetation Shannon index), Tongzhixin (habitat homogeneity index), Yizhixin (habitat heterogeneity index), Soc (soil organic carbon), Phh2o (soil pH), nitrogen (soil total nitrogen), and Ocd (soil coarse debris volume fraction). The impact of these environmental factors on grasshopper community diversity was assessed. Results from redundancy analysis showed that the model’s total explanatory power was 86.38%, with the first and second axes explaining 82.97% and 3.41%, respectively. Significant differentiation in grasshopper community diversity at varying altitudes was observed on the RDA axis. The Shannon–Wiener index, Simpson dominance index, species richness, Pielou index, and Margalef richness index were negatively correlated with the RDA 1 axis, with projection scores of −0.2963, −0.2056, −0.4988, −0.1957, and −0.3448, respectively. The number of individuals exhibited a positive correlation with the RDA 1 axis, with a projection score of 0.3695. The Shannon–Wiener index, Simpson dominance index, and Pielou index showed positive correlations with the RDA 2 axis, with projection scores of 0.0396, 0.0658, and 0.1165, respectively. Conversely, the Margalef richness index, species richness, and number of individuals were negatively correlated with the RDA 2 axis, with projection scores of −0.0407, −0.0796, and −0.0154, respectively. Notably, precipitation in the coldest quarter, precipitation in the driest quarter, and annual average daily range were identified as the environmental variables with the highest absolute projection scores on the RDA 1 axis, scoring −0.7792, −0.7611, and 0.7343, respectively. The vegetation evenness index, annual average daily range, and habitat homogeneity index had the highest absolute projection scores on the RDA 2 axis, scoring −0.4506, 0.3776, and 0.3589, respectively (Figure 7).

Hierarchical segmentation revealed that the total explanation rate of the selected environmental factors for the diversity of grasshopper communities in Xizang was 87.19%, with an unexplainable portion of 12.81%. The environmental factors with the highest explanation rates were the energy factor, water factor, and soil property, accounting for 39.00%, 27.72%, and 14.44%, respectively. Specifically, the three environmental factors with the highest interpretation rates were the annual average daily range, precipitation in the coldest quarter, and precipitation in the driest month, with rates of 9.54%, 9.49%, and 8.89%, respectively (Figure 8A,B).

## 4. Discussion

Insects constitute the most diverse group of animals globally and play a crucial role in biodiversity. Their diversity is vital for maintaining ecological balance [46,47,48,49]. Given Xizang’s unique geography and complex natural environment, studies on insect diversity, particularly concerning grasshoppers, are scarce. In this study, 1159 grasshopper were collected across various altitudinal gradients in Xizang, and 44 species, 28 genera, 11 subfamilies, and 2 families were identified. Among these, *Omocestus cuonaensis* and *Aserratus eminifrontus* were the dominant species. Gomphocerinae was the most abundant family, comprising 534 individuals, accounting for 46.07% of Xizang’s grasshopper population. This dominance probably stems from their broad environmental adaptability and extensive distribution. Furthermore, Oedipodinae exhibited the highest species richness, Margalef richness index, and Shannon–Wiener index, suggesting that Oedipodinae in Xizang forms cohesive communities despite being a dominant group.

The altitudinal gradient can mitigate the confounding effects of species diversity attributable to historical or biogeographic factors and is considered a robust ecological predictor of global climate change dynamics [50,51]. Previous research on species diversity elevation patterns has primarily focused on species richness [52], with comparatively fewer studies examining diverse indices, such as the dominant concentration and evenness indices [53,54]. In this study, we not only evaluated species richness along altitude gradients, but also analyzed patterns in the dominant concentration and evenness indices. The results indicated that grasshopper species richness, Margalef richness index, Shannon–Wiener index, and Simpson dominance index decrease with an increase in altitude in Xizang, which is consistent with the results of Thomas et al. [55]. Their study focused on the altitude distribution model of Orthoptera species richness in the European Pyrenees. König et al. [56] studied the altitude distribution model of Orthoptera species richness in the Central European Alps. The results showed that the species richness consistently decreased with increase in altitude. The decline in grasshopper species richness and Shannon–Wiener index at higher altitudes may be attributed to adverse hydrothermal conditions, high wind speeds, and harsh habitats that limit species survival. Previous studies have explored altitude distribution patterns among various groups, including animals [57,58], microorganisms [59,60], and plants [61] in Xizang, revealing various changes in species richness and biodiversity along altitudinal gradients across different taxa.

Similarities in insect communities primarily reflect similarities in community structure [62]. This study found a similarity coefficient of 0 for the grasshopper community when comparing the altitude ranges of 2600–3100 m and 600–2100 m in Xizang, indicating significant dissimilarity in the community structure between coniferous and broad-leaved mixed forests versus tropical rainforests and evergreen broad-leaved forests. The results also highlighted that differences in plant community structure significantly influence insect community composition. Additionally, as the altitude gradient increases, habitat heterogeneity also increases, leading to considerable differences in grasshopper species, as determined by cluster analysis results. Altitudes of 600–2100 m and 2100–4100 m were clustered into distinct categories. Moreover, the highest similarity coefficient for grasshopper communities was observed when comparing the altitude ranges of 2600–3100 m and 3100–3600 m in Xizang, indicating moderate dissimilarity in community structure, probably due to relatively minor differences in plant community structure between adjacent altitude habitats [63,64].

Altitude variations in temperature and rainfall, influenced by climate change, are key factors shaping insect community diversity and significantly altering species composition within specific ecosystems [65,66]. Studies have found that insects prefer lower temperatures [67], and increasing temperatures may reduce insect diversity [68]. This study also found a significant negative correlation between the Shannon–Wiener index, species richness, Margalef richness index, and Simpson dominance index of the grasshopper community with temperature variables Bio2 (mean diurnal range), Bio4 (temperature seasonality) and Bio7 (annual temperature range). Han et al. [69] found a strong positive correlation between insect species richness and water-related factors such as annual precipitation and precipitation during the warmest season, consistent with the significant positive associations observed in this study between grasshopper species richness and related indices and water factors Bio14 (precipitation of driest month), Bio17 (precipitation of driest quarter) and Bio19 (precipitation of coldest quarter). Previous studies have also confirmed that factors such as temperature and precipitation determine the species richness along the altitudinal gradient. It is generally believed that a decrease in temperature is an important reason for the decrease in diversity with increasing altitude [70,71].

Some studies have found that soil pH can indirectly influence the diversity of phytophagous insect communities by affecting soil fertility composition and the growth, development, species, biomass, and nutrient content of aboveground plants [72,73]. In this study, the Shannon–Wiener index, species richness, Margalef richness index, and Simpson dominance index of the grasshopper community were significantly negatively correlated with soil pH. Conversely, no significant correlation was found between the Pielou index and any soil factors. These findings contrasted with those of Ma et al. [74] concerning grasshopper community diversity in Ningxia grassland, where no significant correlations were found between grasshopper Margalef richness index, Shannon–Wiener index, and soil factors, but a significant positive correlation was observed between the Pielou index and soil bulk density and pH. This disparity may be related to Xizang’s unique geographical location and the broader altitude range covered while sampling. In this study, grasshoppers were investigated at an altitude of 676 to 4100 m. In contrast, Ma et al. [74] examined grasshopper communities in Ningxia grassland within a narrower altitude range of 1219 to 2619 m.

The reasons for forming the altitude distribution pattern of species diversity have been discussed in many aspects. Still, the most relevant environmental factors and the most convincing hypothesis have not yet been found. However, studies have also shown that a single hypothesis cannot perfectly explain the elevational distribution pattern of diversity. Some studies have shown that the habitat heterogeneity hypothesis, plant diversity hypothesis, and water energy dynamic hypothesis can reveal the potential mechanism for the biodiversity distribution pattern [75,76,77]. The results of hierarchical segmentation showed that the total explanation rate of the selected environmental factors to the diversity of the grasshopper community in Xizang was 87.19%, among which the explanation rates of energy factor, water factor, soil property, habitat heterogeneity, and plant diversity were 39.00%, 27.72%, 14.44%, 3.95%, and 2.08%, respectively, indicating that the altitude distribution pattern of grasshopper community diversity in Xizang was driven by the soil property, habitat heterogeneity hypothesis, and plant diversity hypothesis under the guidance of the water energy dynamic hypothesis. Among them, the annual average daily range, the precipitation in the coldest quarter, and the precipitation in the driest month are the three environmental factors that explain the variation in grasshopper community diversity in Xizang.

## 5. Conclusions

In this study, we examined the composition characteristics, diversity distribution pattern, and primary drivers of grasshopper communities across altitudes ranging from 600 to 4100 m in Xizang. Our findings highlighted distinct altitude-related traits in Xizang’s grasshopper communities, where species richness, Margalef richness index, Shannon–Wiener index, and Simpson dominance index decreased as altitude increased. The results of Pearson correlation analysis showed that temperature factors (Bio2, Bio4, and Bio7), water factors (Bio14, Bio17 and Bio19), and soil pH were closely related to the diversity metrics of grasshopper communities along the elevation gradient. Hierarchical segmentation identified the annual average daily range, precipitation in the coldest quarter, and precipitation in the driest month as the top factors explaining variability in grasshopper community diversity in Xizang. These results highlighted the pivotal role of interactions between altitude and the environment in shaping the insect community structure, distribution patterns, and diversity, and these data will form a valuable baseline to assess any future changes resulting from climate change.

## Figures and Tables

**Figure 1 insects-15-00671-f001:**
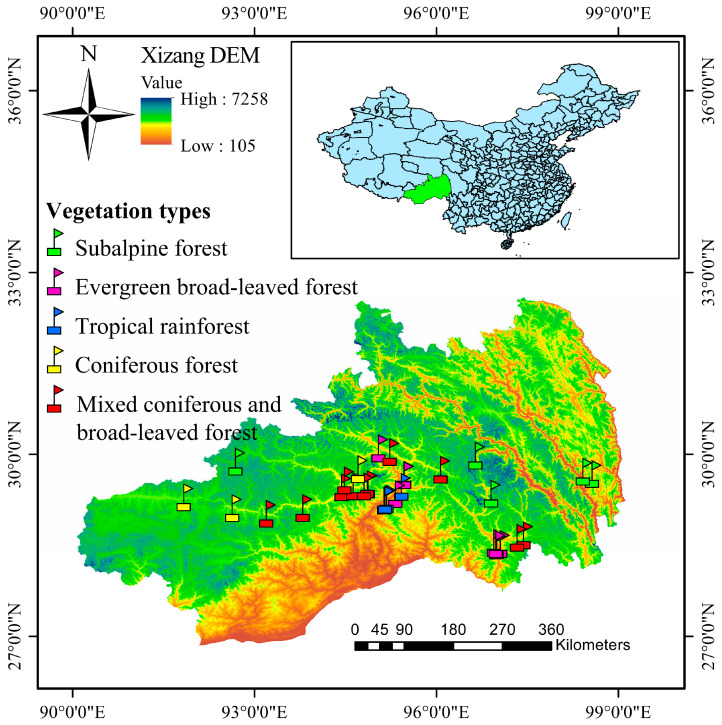
Distribution of sampling points of five vegetation types in the Xizang region.

**Figure 2 insects-15-00671-f002:**
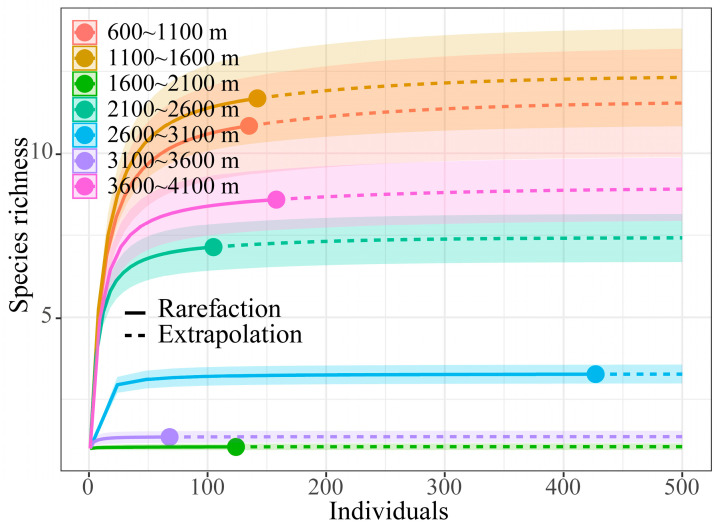
Sparse curve of grasshopper community at different altitudes in Xizang.

**Figure 3 insects-15-00671-f003:**
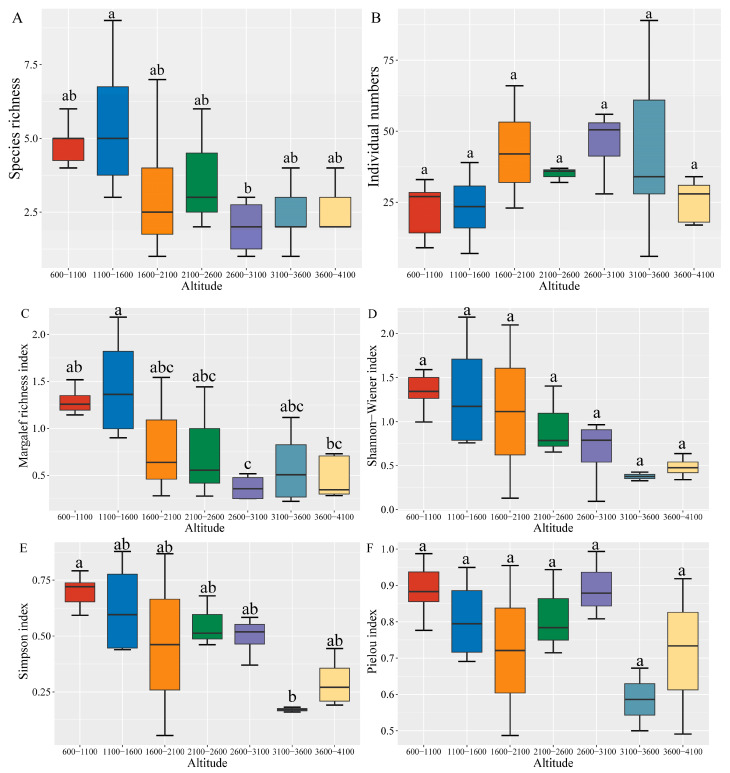
Diversity analysis of grasshopper communities at different altitudinal gradients in the Xizang Region: (**A**) species richness; (**B**) individual numbers; (**C**) Margalef richness index; (**D**) Shannon–Wiener index; (**E**) Simpson index; (**F**) Pielou index. Note: values are mean ± SD of three replicates for each typical vegetation community ecosystem. For each column, values with different letters are significantly different at *p* = 0.05.

**Figure 4 insects-15-00671-f004:**
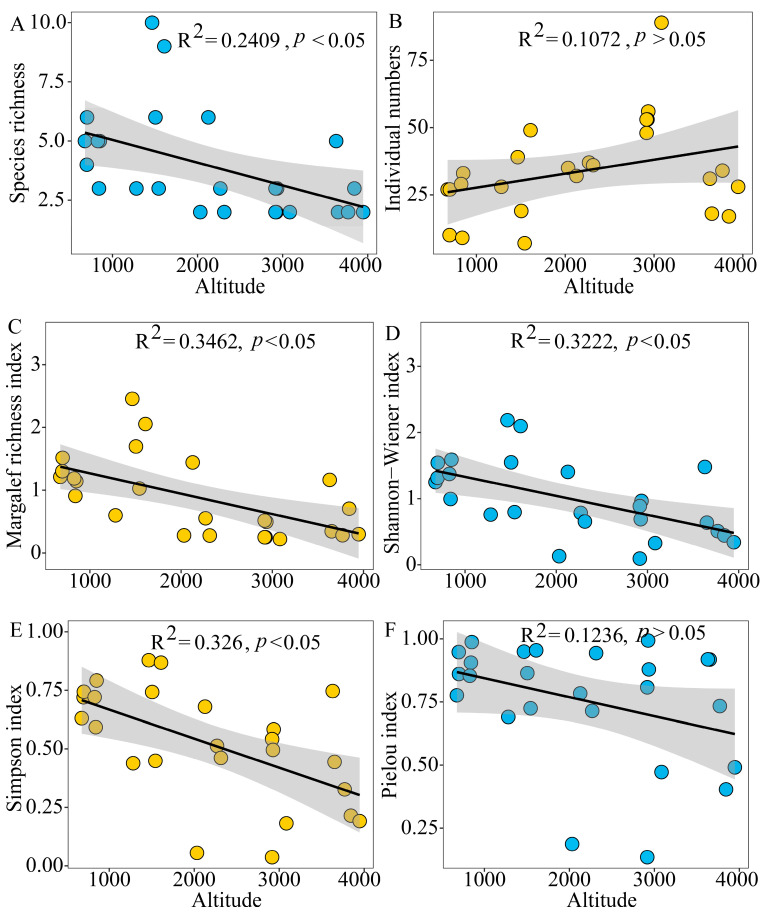
Elevation distribution pattern of grasshopper community diversity in the Xizang region: (**A**) distribution pattern of species richness; (**B**) distribution pattern of individual numbers; (**C**) distribution pattern of Margalef richness index; (**D**) distribution pattern of Shannon–Wiener index; (**E**) distribution pattern of Simpson index; (**F**) distribution pattern of Pielou index.

**Figure 5 insects-15-00671-f005:**
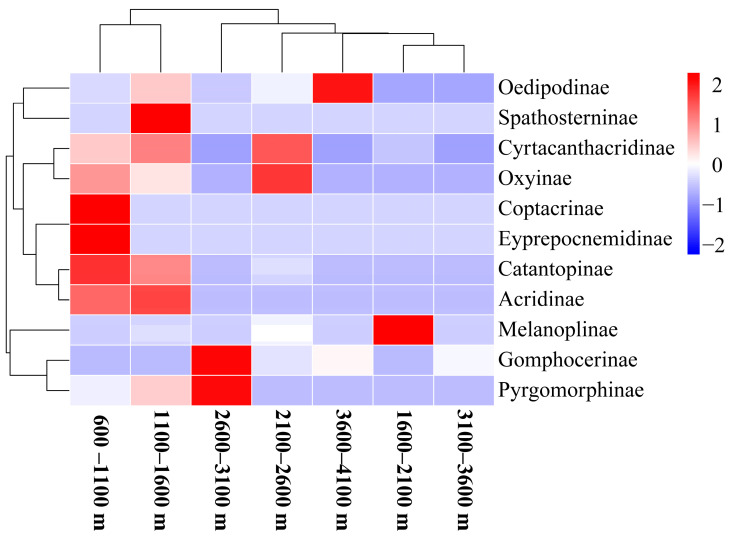
Heat map of the clustering of grasshopper family-level community structure at different altitudinal gradients in the Xizang region.

**Figure 6 insects-15-00671-f006:**
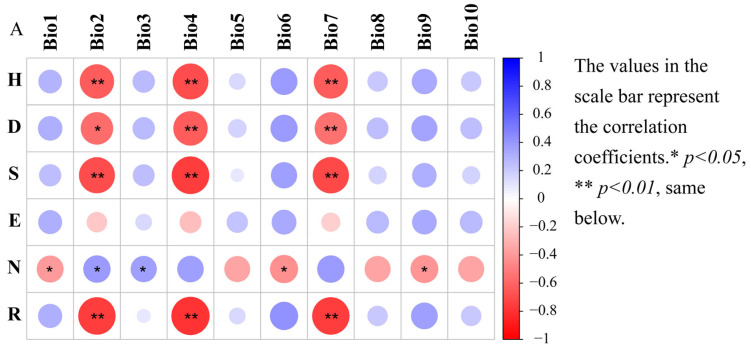
Heat map of the Pearson correlation between grasshopper community diversity and environmental factors at different altitudinal gradients in the Xizang region. Note: (**A**) Correlates with temperature factor; (**B**) Correlates with moisture factor; (**C**) Correlates with vegetation and soil factors. The color of the circle indicates the direction of the correlation, where blue indicates positive correlation and red indicates negative correlation, and the darker the color indicates stronger correlation.

**Figure 7 insects-15-00671-f007:**
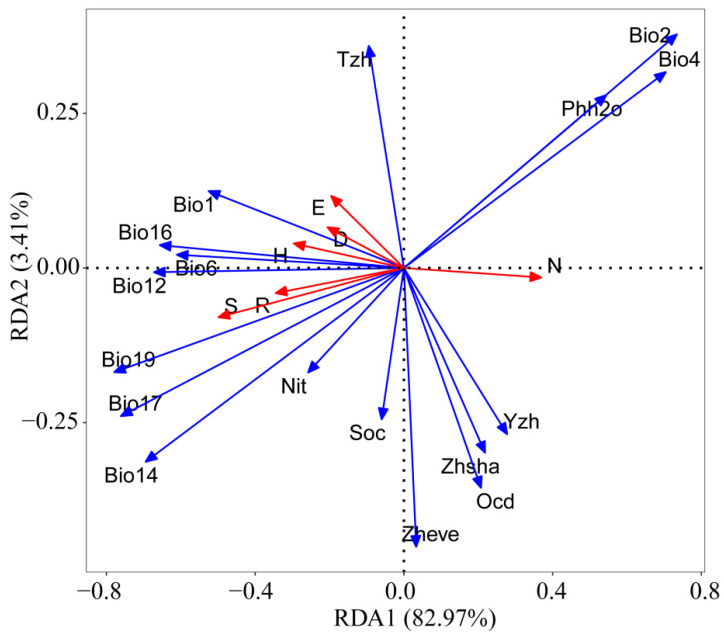
Redundancy analysis (RDA) ranking of grasshopper community diversity indices with environmental factors in the Xizang region.

**Figure 8 insects-15-00671-f008:**
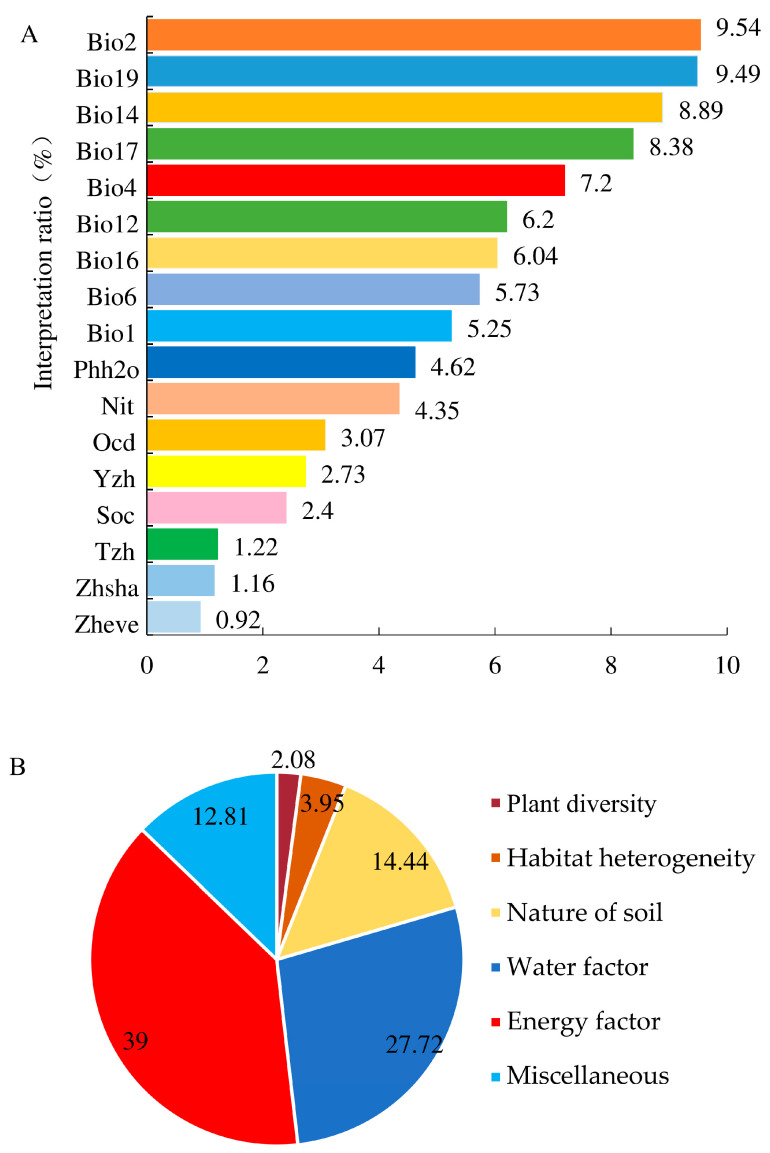
Independent explanatory rates of environmental factors for the grasshopper community diversity in the Xizang region: (**A**) Independent explanatory rate for a single environmental factor; (**B**) Explanatory rates for six categories of environmental factors.

**Table 1 insects-15-00671-t001:** Information on the genera and species of grasshoppers in the Xizang region.

Family	Number of Genera and % of Total Genera Collected	Number of Species and % of Total Species Collected	Ratio Coefficient Genera/Species	Number of Genera with One Species	Number of Genera with > One Species (# of Species)
Gomphocerinae	2 (7.14%)	10 (22.73%)	0.20	0	2 (10 spp.)
Oedipodinae	10 (35.71%)	14 (31.82%)	0.71	6	4 (8 spp.)
Pyrgomorphinae	2 (7.14%)	3 (6.82%)	0.67	1	1 (2 spp.)
Melanoplinae	2 (7.14%)	2 (4.55%)	1.00	2	0 (0 spp.)
Cyrtacanthacridinae	2 (7.14%)	2 (4.55%)	1.00	2	0 (0 spp.)
Catantopinae	3 (10.71%)	4 (9.09%)	0.75	2	1 (2 spp.)
Oxyinae	1 (3.57%)	2 (4.55%)	0.50	0	1 (2 spp.)
Coptacrinae	1 (3.57%)	1 (2.27%)	1.00	1	0 (0 spp.)
Eyprepocnemidinae	1 (3.57%)	1 (2.27%)	1.00	1	0 (0 spp.)
Spathosterninae	1 (3.57%)	1 (2.27%)	1.00	1	0 (0 spp.)
Acridinae	3 (10.71%)	4 (9.09%)	0.75	2	1 (2 spp.)
Total	28 (100%)	44 (100%)	0.64	18	10 (26 spp.)

#: Number of genera with > one species.

**Table 2 insects-15-00671-t002:** Characteristics of the grasshopper community diversity in the Xizang region.

Family	Individual Numbers	Margalef Richness Index	Species Richness	Shannon– Wiener Index	Simpson Dominance Index	Pielou Index
Gomphocerinae	534	1.4330	10	0.6002	0.5146	0.2606
Oedipodinae	148	2.6015	14	1.1685	0.8546	0.4428
Melanoplinae	142	0.2018	2	0.4741	0.2318	0.6840
Pyrgomorphinae	125	0.4142	3	0.6555	0.5791	0.5966
Catantopinae	82	0.6808	4	0.7595	0.5827	0.5479
Oxyinae	31	0.2912	2	0.5860	0.4370	0.8454
Acridinae	30	0.8820	4	0.9165	0.6778	0.6611
Coptacrinae	26	-	1	-	-	-
Cyrtacanthacridinae	18	0.3460	2	0.3046	0.1049	0.4395
Spathosterninae	12	-	1	-	-	-
Eyprepocnemidinae	11	-	1	-	-	-

**Table 3 insects-15-00671-t003:** Similarity analysis of grasshopper communities at different altitudinal gradients in the Xizang region.

Altitude (m)	600–1100	1100–1600	1600–2100	2100–2600	2600–3100	3100–3600
1100–1600	0.24					
1600–2100	0.06	0.06				
2100–2600	0.14	0.19	0.10			
2600–3100	0.00	0.00	0.00	0.08		
3100–3600	0.00	0.00	0.00	0.10	0.40	
3600–4100	0.00	0.04	0.00	0.05	0.14	0.08

**Table 4 insects-15-00671-t004:** Abbreviations for diversity indices and environmental factors.

Factor	Description	Factor	Description
Diversity Indices		Precipitation Factors	
H	Shannon–Wiener index	Bio12	annual precipitation
S	species richness	Bio13	precipitation of wettest month
D	Simpson dominance index	Bio14	precipitation of driest month
E	Pielou index	Bio15	precipitation seasonality
R	Margalef richness index	Bio16	precipitation of wettest quarter
N	number of individuals	Bio17	precipitation of driest quarter
Temperature factors		Bio18	precipitation of warmest quarter
Bio1	annual mean temperature	Bio19	precipitation of coldest quarter
Bio2	mean diurnal range	Soil factors	
Bio3	isothermality	Soc	soil organic carbon
Bio4	temperature seasonality	Phh2o	soil pH
Bio5	max. temperature of warmest month	Nit	soil total nitrogen
Bio6	min. temperature of coldest month	Ocd	organic carbon density
Bio7	temperature annual range	Sand	sand ratio
Bio8	mean temperature of wettest quarter	Pdu	slope
Bio9	mean temperature of driest quarter	Gdu	vegetation coverage
Bio10	mean temperature of warmest quarter	Hsh	soil moisture content
Bio11	mean temperature of coldest quarter		
Vegetation factors			
Zheve	vegetation evenness index		
Zhsim	vegetation Simpson index		
Zhsha	Shannon index of vegetation		
Tzh	habitat homogeneity index		
Yzhi	habitat heterogeneity index		

## Data Availability

The data presented in the study are available in the article.

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
