# Peer review of "Effects of Environmental Factors on the Diversity of Grasshopper Communities along Altitude Gradients in Xizang, China"

_insects, 2024, doi:10.3390/insects15090671_

Round 1

Reviewer 1 Report

Comments and Suggestions for Authors

The manuscript provides a well-structured investigation of the diversity of grasshopper communities along elevational gradients in Tibet, presenting a compelling exploration of the impact of environmental factors on these communities. I emphasize that the study focuses on an understudied region, which together with the methodology makes it an important contribution to the study. 

As a suggestion to the authors, I would like to comment on Figure 1, which although it effectively illustrates the distribution of sampling points among different types of vegetation in Tibet, it would be beneficial to include a reference map showing the location of the study area. within a global context. This addition would improve the reader's understanding of the geographic setting.

Author Response

Thank you for the opportunity to revise our manuscript titled “Effects of environmental factors on the diversity of grasshopper communities along altitude gradients in Xizang, China”(insects-3141126). We appreciate the valuable feedback from you  that has greatly improved our work. 
Below are our responses to  your’s comments:

Comments: The manuscript provides a well-structured investigation of the diversity of grasshopper communities along elevational gradients in Tibet, presenting a compelling exploration of the impact of environmental factors on these communities. I emphasize that the study focuses on an understudied region, which together with the methodology makes it an important contribution to the study.

As a suggestion to the authors, I would like to comment on Figure 1, which although it effectively illustrates the distribution of sampling points among different types of vegetation in Tibet, it would be beneficial to include a reference map showing the location of the study area. within a global context. This addition would improve the reader's understanding of the geographic setting.

Response: The comments were reviewed, and following the teacher's input, the issue was discussed among the group authors. It was concluded that the location of the study area is highlighted in green on the map of China located in the upper right corner, ensuring that the study area's location is clearly visible.

Reviewer 2 Report

Comments and Suggestions for Authors

Major revisions required as often I found it difficult to follow the results.  

Author Response

Thank you for the opportunity to revise our manuscript titled “Effects of environmental factors on the diversity of grasshopper communities along altitude gradients in Xizang, China”(insects-3141126). We appreciate the valuable feedback from you  that has greatly improved our work. 
Below are our responses to  your’s comments:

Comments 1: “Line 36-38: «..Margalef richness of grasshopper communities peaked at 1100-1600, and then declinedsignificantly (P < 0.05) with increasing altitude."

Response 1: Thank you for the reminder. The revision based on the reviewers' comments is as follows (Line 36-38): “The results indicated a significant decrease (P < 0.05) in species richness and the Margalef richness index of grasshopper communities with increasing altitude, with the highest values observed at 1100–1600 m and the lowest at 2600–3100 m.”

Comments 2: Line 116: "Tibet covers 1.2 million km', which is roughly one- eighth.. using so many significantfigures is not consistent with saying "roughly one- eighth

Response 2: Thank you for the reminder. The revised statement based on the reviewers' comments is as follows (Line 114): “Tibet spans an area of 1.202189 million km2.”

Comments 3: Line 119-122: ,,Tibet alpine area. Moving from northwest to southeast, the plateau transitions from and cold, dry temperate climate to a warmer, more humid subtropical and finally tropical climate. Theaverage...

Response 3: We agree with the comments. Changes have been made in the article where the description was deemed inappropriate (Line 116-118).

Comments 4: Line 128: ".rom August to October 2020-2022."

Response 4: Thank you for the reminder. The revised statement based on the reviewers' comments is as follows (Line 125): “Tibet from August to October 2020-2022.”

Comments 5: Line 132: Within each quadrat, grasshoppers were collected by sweep net, with surveyors movingalong both diagonals of the quadrat."

Response 5: We agree with the comments. Changes have been made in the article where the description was inappropriate (Line 129-130).

Comments 6: Line 137-141: The collections were made in the central south of the Tibetan plateau in a narrow rangeof latitudes where lower levels were warm humid and higher levels were cold. Provide the range oflatitudes of the collecting sites eg 28-32°N

Response 6: We agree with the comments. The original writing caused some ambiguity, and it has now been revised as follows based on the reviewers' comments (Line 135-137): “Collections were conducted in the central southern region of the Tibetan plateau within a narrow latitude range (28-31°N), where the lower elevations were characterized by warm, humid conditions and the higher elevations were cold.”

Comments 7: Section 2.4.2 Similarity analysis: use the term Jaccard similarity" as you do in line 286 of the results.

Response 7: We agree with the comments. Specific changes have been made in the corresponding sections of the article. (Line 185 )

Comments 8: Lines 208-214: "A total of 1159 grasshoppers were collected in Tibet with Gomphocerinae being themost common (534/1159 = 46.1%) followed by Oedipodinae (148 or 12.8%), Melanopinae (12.3%) and Pyrgomorphinae (10.8%), with remaining seven subfamilies making up 18.0%. Omocestus cuonaensisand Aserratus eminifrontus were the dominant species, representing 30.03% and 10.26% of the total grasshoppers collected.

The grasshoppers collected were in two families, 11 families, 28 genera and 44 species. Of these. 18 genera (64.3%) were represented by only one species, demonstrating a predominance ofsingle-species generera among the grasshoppers we collected from Tibet. The ration coefficient ofgenera to species was 0.64. Subfamilies ...

Table 1 is difficult to read I suggest something like:

Table 1. Information on the genera and species of grasshoppers collected in Tibet.

Response 8: We agree with the comments. Specific changes have been made in the corresponding sections of the article, and the data table has been reintegrated. (Line 221-231)

Comments 9: Lines 223-234:“The relative frequency of grasshoppers collected are given in Table 2, with Gomphocerinaemost common and Erprepocnemidinae least common. Margalef richness was highest in Oedipodinae followed by Gomphocerinae ..... Species richest was highest in Oedipodinae followed by.... etc.

Response 9: The comments are acknowledged. Specific modifications have been implemented in the relevant sections of the article. (Line 235-246)

Comments 10: Line 248: “...reaching a maximum at 1100- 1600m and a minimum at 2600-3100 m (Fig 2A, 2C).”

Response 10: Based on the comments, the amendment has been made as follows (line 267-268) : reaching a maximum at 1100–1600 m and a minimum at 2600–3100 m (Fig. 3A, 3C).

Comments 11: Line 251-253: The Pielou index does look to be highest at 600-1 100m and lowest at 3100-3600m yet in Fig2F all of them are shown as not significantly different as all have“a”. Similar with Line 253-254: The number of individuals does look higher at 2600-3 100m but why are they marked as“a”in fig 2B whichmeans NOT significantly different. The data in Figure 2 need further explanation.

Response 11: The comments are acknowledged. Based on the teacher's feedback, the Shannon-Wiener index, Pielou index, and number of individuals were re-analyzed. The results indicated that none of the three indices showed significant differences across the altitudinal gradients. The maximum and minimum values described in the article were based on the mean size. (line 264-265)

 Comments 12: Figure 3: Where in the text do you refer to Figure 3? Should this be in lines 247-250?

Response 12: The comments are acknowledged. The corresponding part of the article has been labeled with Figure 3. (line 266-274)

Comments 13: Line 275:“Figure 4 shows that Catantopinae...

Response 13: The comments are acknowledged. Specific changes have been made in the relevant sections of the article. (line 292)

Comments 14: Line 288-296: “The highest similarity coefficient of 0.40 was observed between the adjacent 2600-3 100mand 3 100-3600m altitudes but even this level demonstrated moderate dissimilarity between grasshopperspresent at these altitudes. Highly dissimilar relationships of 0.14 to 0.24 were observed between four otheraltitudes but 8 of the 21 relationships had no overlap in species at all, reflecting the overall dissimilarity ofgrasshoppers at various altitudes (Table 3).

Response 14: The comments are acknowledged. Specific modifications have been made in the relevant sections of the article. (line 305-310)

Comments 15: Figure 5 and text in lines 300-325: It is very difficult to remember what each of the Bio factors are.Therefore, I suggest for line 305+:“Significant negative correlations were found between the Shannon-Weiner index (Fig. 5:H) of the grasshopper community and factors such as soil pH (Fig 5: Phh2o), annualaverage daily range (xx), temperature seasonal variation (yy).." Tell us what you are referring to in Fig.5. I suggest before Figure 5 you list all of the Bio factors in a Table so that they are easy to find for Figures 5-7. In the written list in lines 332-345 is not easy to find each Bio.

Response 15: The comments are acknowledged. All diversity indices and abbreviations of environmental factors have been listed in Table 1. (line 217)

Comments 16: Line 398:“in this study 1,159 grasshoppers were collected across various altitudina...

Response 16: The comments have been acknowledged. Specific modifications have been implemented in the relevant sections of the article. (line 403)

Comments 17: Line 416-417:“..results ofThomas et al. [54] whose study focused on the altitude distribution..."Line 420:“..species richness consistently decreased with increase in altitude."

Response 17: The comments have been agreed upon. Specific changes have been made in the relevant sections of the article. (line 421-422)

Comments 18: Line 447: include what Bio2, Bio4 and Bio7 are here. Same in lines 451-452.

Response 18: The comments have been accepted. Specific changes have been made in the appropriate sections of the article. (line 457, 452)

Comments 19: Line 499-501:“These results highlighted.... .structure, distribution patterns and diversity and these datawill form a valuable baseline to assess any future changes resulting from climate change.

Response 19: Based on the reviewer's comments, it has been amended as follows ((line 504-507 ) : “These results emphasize the crucial role of interactions between altitude and the environment in determining insect community structure, distribution patterns, and diversity. These data will serve as an important baseline for evaluating future changes due to climate change.”

Reviewer 3 Report

Comments and Suggestions for Authors

My suggestions have been attached.

Author Response

Thank you for the opportunity to revise our manuscript titled “Effects of environmental factors on the diversity of grasshopper communities along altitude gradients in Xizang, China”(insects-3141126). We appreciate the valuable feedback from you  that has greatly improved our work. 
Below are our responses to  your’s comments:

Major Comments:

Comments 1: The authors used three years of field data, but can these data represent the grasshoppers completeness in the study area? In other words, how much influence does sampling bias have on the diversity pattern of the grasshoppers community in this study? It is well known that when studying species diversity patterns, it is necessary to evaluate sampling bias. The authors can add a species cumulative curve analysis to evaluate the impact of sampling bias on grasshoppers diversity patterns, which is easy to implement in R software packages (e.g., vegan package).

Response 1: The comments have been agreed with, and based on this, sparse extrapolation curves for species with varying elevation gradients have been added to the article to demonstrate the adequacy of sampling for locust communities. (line 257-259 )

Comments 2: Spatial autocorrelation may lead to bias in model parameter estimates and can improve type I error rates. Therefore, I suggest that authors should consider spatial autocorrelation when using regression and correlation analysis to evaluate the relationship between grasshopper diversity and environmental variables. Spatial autocorrelation can be used in SAM and R software. The following literature may be helpful to you.

Dormann, C. F., McPherson, J. M., Araujo, M. B., Bivand, R., Bolliger, J., Carl, G., … Wilson, R. (2007). Methods to account for spatial autocorrelation in the analysis of species distributional data: A review. Ecography, 30, 609–628.

Rangel, T.F.; Diniz–Filho, J.A.F.; Bini, L.M. SAM: A comprehensive application for spatial analysis in macroecology. Ecography 2010, 33, 46–50.

Kissling, W.D., Carl, G., 2008. Spatial autocorrelation and the selection of simultaneous autoregressive models. Glob. Ecol. Biogeogr. 17, 59–71.

Response 2: The comments are agreed with. The original writing did lead to ambiguity, and it has now been revised as follows based on the reviewers' comments ((Line 205-209) “Due to the significant spatial autocorrelation typically present in species diversity and environmental data, regression and correlation analyses often show increased significance. To address the impact of substantial spatial autocorrelation on the significance of results, the results of regression and correlation analyses are evaluated using a modified t-test.”

Comments 3: Why consider the collinearity of environment variables before redundancy analysis (RDA)?

Response 3: Covariance can complicate the evaluation of the impact of individual environmental variables, making it challenging to clearly identify the independent role of each variable. Additionally, filtering by software may result in the removal of critical factors, such as Bio1 and Bio12, which are important for explaining the elevation pattern of locust diversity. Consequently, prior to conducting RDA analysis, appropriate screening of moisture and temperature factors was performed, referencing previous studies for guidance. (Line 356-382)

Comments 4: As described in the manuscript, Xizang's unique geographical location makes it an ideal region for studying community diversity. Under this logic, there should be many studies on the altitude patterns of insects, but I did not see the authors compare the results of this study with them in the discussion section. I suggest adding this point. In addition, although the authors obtained the main environmental variables, they did not provide an in-depth explanation of how these environmental variables shaped the altitude pattern of grasshopper in Xizang. Furthermore, Introduction part state “this study aims to provide a framework for the conservation of insect diversity in Tibet” (Line 110-111). However, this is not mentioned in the Discussion section. I suggest adding this.

Response 4: The comments are agreed with. A comprehensive review of references has been conducted. Xizang's unique geographical location makes it an ideal region for studying community diversity. However, while studies on altitudinal patterns of diversity have predominantly focused on microorganisms, animals, and plants, research on altitudinal patterns of insect diversity is limited, with only a few studies addressing this topic. Furthermore, a detailed explanation of the relationship between environmental factors and grasshopper diversity in Xizang has been provided in the Discussion section, with views on how these environmental variables influence the altitude pattern of grasshoppers in Xizang discussed between lines 482-492. .

Minor Comments:

Comments 1: Tibet is an old name, please replace it with Xizang in entire manuscript.

Response 1: Thank you for the comments. All instances of 'Tibet' have been replaced with 'Xizang' in the article.

Comments 2: Line 45-46:removing the Bio2, Bio4, Bio7, Bio12~Bio14, and Bio16~Bio19. 

Response 2: The comments are agreed with. Bio2, Bio4, Bio7, Bio12 through Bio14, and Bio16 through Bio19 have been removed from the corresponding sections of the article.  (line 45-46)

Comments 3: Line 58-59: “where there are plants, there will be grasshoppers”. This sentence is too absolute and needs to be revised.

Response 3: Based on the reviewer's comments, it has been amended as follows ((line57-58): Grasshoppers inhabit diverse environments and exhibit high adaptability to varying conditions.

Comments 4: Line 131:Why selected 10 m x 10 m quadrats? 

Response 4: Thank you for the question. This issue is critical for the study of locust diversity. In response, all authors discussed and concluded that a sample size of 10 m x 10 m is suitable for Xizang. This size is neither too large to cause operational difficulties nor too small to accurately reflect population density. A 10 m x 10 m sample size can be consistently maintained across each plot, ensuring uniformity in the surveyed area. Appropriate sample sizes help mitigate the impact of local environmental changes, making the samples more representative of locust distribution throughout the study area.

Comments 5: Line 135-136:What tools are used to measure latitude, longitude, altitude, soil

moisture content, and slope? Please provide.

Response 5: Based on the reviewer's comments, it has been amended as follows (line 130-133) : The longitude, latitude, elevation, and slope of each sample plot were recorded using a hand-held GPS unit and compass; species, cover, number of plants (clumps), and height within each grassland sample plot were assessed.

Comments 6: Line 138: How many sampling points are set for each altitude gradient? I didn't get that information.

Response 6: Thank you for the feedback. The number of sampling points has been included in the article. (line 137-141)

Comments 7: Line 138-141:“Based on the vegetation types observed in the area, .........subalpine forest (3600 ~ 4100 m)(Fig.1)”. This sentence needs to be reorganized, I propose to change it as: Based on the five vegetation types observed in the area (Fig.1), the region was categorized into seven elevation gradients:600 ~ 1000 m, 1100 ~ 1600 m, 1600 ~ 2100 m, 2100 ~ 2600 m, 2600 ~ 3100 m, 3100 ~ 3600 m, and 3600 ~ 4100 m).

Response 7: Thank you for the comments. Specific changes have been made in the article as follows: “The region was categorized into seven elevation gradients based on the five vegetation types observed in the area (Fig. 1): 600 ~ 1000 m (setting 6 sample points), 1100 ~ 1600 m (setting 5 sample points), 1600 ~ 2100 m (setting 3 sample points), 2100 ~ 2600 m (setting 3 sample points), 2600 ~ 3100 m (setting 8 sample points), 3100 ~ 3600 m (setting 3 sample points), and 3600 ~ 4100 m (setting 5 sample points).” (line 136-141)

Comments 8: Please add altitude legend in Fig 1, and the color of the study area in the map of China should be consistent with the color of the picture below. In addition, the resolution of the Fig 1 is low.

Response 8: The comments are agreed with, and the following revisions have been made: an altitude legend has been added to Figure 1, the colors of the study area on the China map have been changed and a higher resolution image has been provided. (line 145)

Comments 9: Line 146:Please indicate the year period of 19 environmental factors.

Response 9: Thank you for the comment. The year of the environmental factor has been provided in section 2.3.1. (line 150)

Comments 10: Line 147: Is it v1.4 or 2.1? Please confirm.

Response 10: Thank you for the comment. It has been verified as v 2.1. (line 150)

Comments 11: Line 147:In 2.3 Environmental data part, what is the resolution of the environmental raster data? Please provide.

Response 11: Thank you for the comment. The spatial resolution for extracting data for the three types of environmental factors has been listed in section 2.3 of the paper. (line 150,162,174)

Comments 12: Line 165-167:The full name should be written first and then the corresponding abbreviation in brackets, e.g., soil organic carbon (SOC).

Response 12: Thank you for the feedback. In response to your comments, revisions have been implemented in the relevant sections of the article. (line 169)

Comments 13: Line 187-189:“highly dissimilar when Cs ranges from 0 to 0.25, moderately dissimilar when Cs ranges from 0.25 to 0.50, moderately similar when Cs ranges from 0.50 to 0.75, and highly similar when Cs ranges from 0.70 to 1.00”, Please add cited references.

Response 13: Thank you for the comment. References have been cited at the end of this paragraph. (line 194)

Comments 14: Line 191:delete “The data were collated using Microsoft Excel 2019”.

Response 14: Thank you for the comment. The statement “The data were collated using Microsoft Excel 2019” has been removed. (line 196)

Comments 15: Line 208-209:The manuscript states“A total of 1,159 grasshoppers were collected in Tibet, categorized into two families, 11 subfamily, 28 genera, and 44 species”. Since there are many different classification systems for grasshoppers, please indicate which classification system you use in the methods section. In addition, replace "two" with "2".

Response 15: Thank you for the comment. Specific grasshopper classification systems have been noted in the article. (line 133)

Comments 16: Line 213:I not obtained the 30.03% and 10.26% in Table 1.

Response 16: Thank you for the comment. Since 30.03% and 10.26% represent the abundances of the two dominant species, it is not feasible to include abundance tables for all species in the article. However, species data for all sampling sites are available at the provided link. (line 509)

Comments 17: Line 208-220:The data described in the manuscript cannot be directly obtained from the Table 1. I suggest remaking the Table 1.

Response 17: Thank you for the comment. Table 1 has been revised. (line 232)

Comments 18: Line 225, 233: the part “Coptacrinae > Cyrtacanthacridinae > Spathosterninae > Eyprepocnemidinae”;“Melanoplinae > Acridinae > Pyrgomorphinae > Catantopinae > Oedipodinae > Cyrtacanthacridinae > Gomphocerinae”. The font size should be consistent with the rest of the font in the section

Response 18: Thank you for the comment. The font size in that section of the article has been adjusted. (line 235-246)

Comments 19: Significant P values are in italics, please check the full text

Response 19: Thank you for the comment. Full-text checks have been performed, and significant P values are now shown in italics. (full text)

Comments 20: Most of the figure in the manuscript are of low resolution, e.g., figure 1, 2, 5 

Response 20: Thank you for the comment. High-resolution images have been included in the article, and high-resolution TIFF images are provided in the attachment (Figures 1, 2, 5).

Comments 21: Line 416-417:“….results of Thomas et al. 416[54]Their study…..”replaced as

“….results of Thomas et al. 416[54]. Their study…..” 

Response 21: Thank you for the comment. Specific changes have been made in the article. (line 421-423)

Comments 22: Line 438:delete the cited reference “[62]”

Response 22: Thank you for the comment. The cited reference “[62]” has been deleted. (line 443)

Comments 23: Line 396-406:These sentences are just repetitive descriptions of the research results and do not need to be included in the Discussion section. I suggest deleting them.

Response 23: The comments are acknowledged. Following the teacher's opinion, the authors discussed the potential removal of this section and concluded that it is essential to the article. Removing it could affect the structural integrity of the manuscript. (line 400-411)

Comments 24: There are many incorrect uses of reference format throughout the manuscript,e.g.,

Line 56 [3-4] and Line 83 [28-29]. Correct use: [3, 4],  [28, 29]

Response 24: Thank you for the comment. The reference has been correctly quoted in the article.

Comments 25: Line 99-101:delete the sentence “The Qinghai-Tibet Plateau....................... across

the region ”.

Response 25: Thank you for the comment. The sentence “The Qinghai-Tibet Plateau....................... across the region” has been deleted. (line 98)

Round 2

Reviewer 2 Report

Comments and Suggestions for Authors

This paper is much improved with a few minor changes suggested:               Line 314-ff: "Six dependent variables (Shannon-Wiener index (H), species richness (S), Simpson dominance index (D), Pielou index (E) and the number of individuals (N) were correlated with 32 independent variables (Table 1).     Line 451-ff: "..with temperature variables Bio2 (mean diurnal range), Bio4 (temperature seasonality) and Bio7 (annual temperature range)."                            LIne 455-ff: "...water factors Bio14 (precipitation of driest month), Bio17 (precipitation of driest quarter) and Bio19 (precipitation of coldest quarter)."

Reviewer 3 Report

Comments and Suggestions for Authors

Dear author, thank you for accepting my suggestions. Now I have no questions about the manuscript.